# Enhanced Tooth Region Detection Using Pretrained Deep Learning Models

**DOI:** 10.3390/ijerph192215414

**Published:** 2022-11-21

**Authors:** Mohammed Al-Sarem, Mohammed Al-Asali, Ahmed Yaseen Alqutaibi, Faisal Saeed

**Affiliations:** 1College of Computer Science and Engineering, Taibah University, Medina 42353, Saudi Arabia; 2Department of Computer Science, Sheba Region University, Marib 14400, Yemen; 3Department of Prosthodontics and Implant Dentistry, College of Dentistry, Taibah University, Al Madinah 41311, Saudi Arabia; 4Department of Prosthodontics, College of Dentistry, Ibb University, Ibb 70270, Yemen; 5DAAI Research Group, Department of Computing and Data Science, School of Computing and Digital Technology, Birmingham City University, Birmingham B4 7XG, UK

**Keywords:** pretrained deep learning, missing teeth, CBCT, DenseNet169 model, CNNs, image segmentation, U-Net model

## Abstract

The rapid development of artificial intelligence (AI) has led to the emergence of many new technologies in the healthcare industry. In dentistry, the patient’s panoramic radiographic or cone beam computed tomography (CBCT) images are used for implant placement planning to find the correct implant position and eliminate surgical risks. This study aims to develop a deep learning-based model that detects missing teeth’s position on a dataset segmented from CBCT images. Five hundred CBCT images were included in this study. After preprocessing, the datasets were randomized and divided into 70% training, 20% validation, and 10% test data. A total of six pretrained convolutional neural network (CNN) models were used in this study, which includes AlexNet, VGG16, VGG19, ResNet50, DenseNet169, and MobileNetV3. In addition, the proposed models were tested with/without applying the segmentation technique. Regarding the normal teeth class, the performance of the proposed pretrained DL models in terms of precision was above 0.90. Moreover, the experimental results showed the superiority of DenseNet169 with a precision of 0.98. In addition, other models such as MobileNetV3, VGG19, ResNet50, VGG16, and AlexNet obtained a precision of 0.95, 0.94, 0.94, 0.93, and 0.92, respectively. The DenseNet169 model performed well at the different stages of CBCT-based detection and classification with a segmentation accuracy of 93.3% and classification of missing tooth regions with an accuracy of 89%. As a result, the use of this model may represent a promising time-saving tool serving dental implantologists with a significant step toward automated dental implant planning.

## 1. Introduction

The placement of dental implants to restore missing teeth is now recognized as a standard dental procedure [1]. A priori dental implant planning is required to achieve good treatment results because it helps to find the correct position of the implant and eliminates surgical risks [2,3,4]. Generally, an implant placement plan is created using a patient’s cone beam computed tomography (CBCT) image, which offers three-dimensional (3D) information to surgeons instead of traditional methods [5]. Finding missing teeth regions missing tooth areas is a critical step that the oral implantologist should determine during implant placement planning [6,7].

With the widespread use of dental implants nowadays, automatic detection of missing tooth areas is necessary to develop an automatic implant plan. Unfortunately, due to the lack of automated technologies, clinicians had to plan to place dental implants manually, which is not efficient as it requires more time and effort [6,7]. Therefore, several studies utilized the power of A.I. to improve the effectiveness and efficiency of clinical treatment methods. In contrast to human intelligence, A.I. is known as machine intelligence [8], where the machines can simulate human minds in the ability to learn and analyze and then solve problems [9,10]. As A.I. technologies have rapidly developed, they has been widely used in several fields, including healthcare. These technologies include the Internet of Things (IoT), machine learning, deep learning, natural language processing, and robotics. The A.I. technologies are successfully applied in the healthcare industry to improve clinical treatment outcomes, enhance the performance of the decision-making process, and enhance clinical diagnosis in various medical task domains [10,11].

One of the main types of machine learning based on artificial neural networks is deep learning (DL), which uses multiple layers of processing to extract knowledge and insight from the data. DL has been applied for several medical applications, such as computer-aided diagnosis and surgical planning, disease classification, tumor detection, and anatomic modeling [12]. In dentistry, it has been applied for tasks such as third-molar extraction, caries detection, prediction of extraction difficulty for mandibular third molars, mandibular canal detection, and tooth numbering [13,14,15,16].

This paper further extends the research of applying artificial intelligence and deep learning in dentistry. It aims to enhance the 3D missing tooth area planning using pretrained deep learning models by automatically detecting and classifying the CBCT images and passing only those which need further planning.

## 2. Related Works

Nowadays, healthcare is one of the basic needs of all people, and smart healthcare has become essential to provide efficient services for several healthcare sectors that improve the quality of people’s lives. Different advanced technologies have been recently used in smart healthcare, which includes artificial intelligence (A.I.), machine learning (ML), deep learning (DL), Internet of Things (IoT), big data analytics, medical sensors, and others [17]. For instance, the traditional diagnosis methods for conditions such as brain, heart, kidney, and liver diseases were manual and error-prone. Unlike exclusive human expertise, applying artificial intelligence methods provides auto diagnosis and reduces detection errors [18]. In dentistry, artificial intelligence has been recently applied to provide clinicians with high-quality patient care and prediction tools that simplify complicated diagnosis protocols. For instance, deep learning methods can effectively detect caries using radiographs as the input layer for these DL methods. Based on A.I. training, multiple hidden layers of DL methods can be used to classify the radiographs into “caries” and “no-caries” [19].

Another important application of A.I. in dentistry is the automated detection of teeth and small edentulous regions. In these applications, cone beam computed tomography (CBCT) images of different patients (such as fully and partially dentate patients) were used for this purpose. Gerhardt et al. [20] used the A.I. tool (Virtual Patient Creator, Relu BV, Leuven, Belgium) to train and test the model with 175 CBCT images. This A.I. tool was used to label, segment, and detect the missing teeth. Based on the 3D U-Net architecture, a deep learning method was used in this tool for detecting and segmenting the teeth in two subsequent networks. The teeth were detected in the down-sampled image, and rough segmentation was provided for the first network. The regions of interest (R.O.I.) can be proposed using the rough segmentation that enabled the use of deep learning for multi-class classification after cropping and down-sampling the images to a fixed resolution. The automated tooth segmentations for all CBCT scans provided by this tool in the form of individual 3D models were judged by an expert. If adjustments were needed, the tooth segmentations were manually refined using this tool.

Similarly, Zhang et al. [21] and Chen et al. [22] conducted studies on automated detection/labeling of 2D teeth, where CNN was applied to detect teeth in periapical radiographs. The experimental results of these two studies showed that the obtained precision rates were 95% and 90%, respectively. These findings ensured the importance of A.I. in providing accurate and efficient detection for proper surgical and treatment planning and automated dental charting.

In [23], Liu et al. used an A.I.-based system to design the implant plan automatically. They applied transfer learning using A.I. to guide implant placement in the posterior mandible. These systems are known as single-shot multibox detectors (S.S.D.) and voxel-to-voxel prediction network for pose estimation (V2V-PoseNet). The characteristic annotators were detected, and the implant position was determined using this A.I. system. In this study, the CNN model was trained using a dataset of 2500 different edentulous sites for CBCT images. The experimental results showed that the A.I. system could use deep learning to effectively detect the anatomy of the object region and generate the ideal implant plan.

Similarly, in [2], Kurt Bayrakdar et al. used an AI-based system for implant planning. A dataset of 75 CBCT images was used in this study to train the deep convolutional neural network for providing an automatic detection of canals/sinuses/fossae linked with alveolar bones and missing tooth regions. The voxel-perfect segmentations were required for predictions of crucial implant planning. The segmentation of missing teeth depends on the segmentations of present teeth and jaws. Using the neighboring teeth location, tilt, and placement, the missing tooth mask was extracted that can be used as a guide to a slicing algorithm and helps predict a mesiodistal angle of implant placement. The outcomes of A.I. methods were compared with the manual assessment using the Wilcoxon signed-rank test and Bland–Altman analysis. The experimental results showed no significant difference between these two methods for the bone height measurements obtained. However, it was significant for bone thickness measurements in all regions of maxilla and mandible. The detection accuracy obtained in this study was 95.3% for missing tooth regions, 72.2% for canals, and 66.4% for sinuses/fossae.

Deep learning methods have recently been used for dental implant planning to detect missing tooth regions. A dataset of 455 panoramic radiographic images was used in [6] to annotate, segment, and detect the missing tooth regions. The tooth masks were generated first during the image segmentation process based on the instance-by-instance method. Mask R-CNN and ResNet-101 were used for the segmentation model. For the detection of missing tooth region, the Faster R-CNN model was used. The backbone of this model was ResNet-101. The experimental results showed that the detection model obtained an average of 92.14% for tooth instance segmentation and 59.09% for missing tooth region detection. In addition, deep learning has also been applied to perform different tasks in dentistry, which include dental caries detection [24] and prediction of extraction difficulty for mandibular third molars [25].

In this paper, several pretrained deep learning models were applied for enhancing 3D missing tooth area planning, including the AlexNet, VGG16, VGG19, ResNet50, DenseNet169, and Mo-bileNetV3 models.

## 3. Materials and Methods

### 3.1. Study Design

In this study, deep learning models were trained to detect and classify the missing teeth region from a dataset segmented from CBCT images of confidential patients who regularly visited the Taibah University dental hospital. The experimental part of this study was implemented jointly at the Faculty of Dentistry and the College of Computer Science and Engineering at Taibah University, K.S.A. Generally, the proposed approach is divided into three main stages: the data acquisition stage, the data preprocessing and segmentation stage, and the feature extraction and classification stage. To ensure that the proposed model was able to classify the images correctly, the collected images were of different patients of different ages. As a result, 500 CBCTimages were collected. Since the images were in 3D dimension, an expert with more than 10 years of professional experience in dental implant planning was to be responsible for slicing and extracting 2D X-ray images from the original CBCT images. The professional dentist was also asked to manually annotate the sliced/extracted images as well as manually segment the missing tooth regions and positions. As a result, the obtained images were used as ground-truth masks for training the proposed deep learning models. In the second stage, the dataset along with the masks, were fed to the segmentation stage, in which augmentation was also performed. In this study, we designed two experiment types based on the presented segmentation techniques. For this purpose, the U-Net model was used in the segmentation stage. In terms of classification models, several deep learning architectures were implemented and tested, including the AlexNet, VGG16, VGG19, ResNet50, DenseNet169, and MobileNetV3 models. These models were used to classify the resulting binary masks from the segmentation stage well as to classify the original images without segmentation. As the pretrained DL models are originally capable of classifying up to 1000 different classes, the last fully connected layer of these models was replaced so that it can be capable of classifying the binary classes of the used dataset. We also compared the output of the proposed models to the ground-truth dataset that was annotated by the domain expert. Figure 1 shows the whole process of the followed methodology.

### 3.2. Ethics Statement

Generating a real dataset of images, which are more than confidential patients’ CBCT images representing different missing tooth positions of patients with different age ranges, the CBCT images were officially obtained from patients between 16 and 72 years of age. The research ethics committee, College of Dentistry, Taibah University, Madinah, Saudi Arabia, approved this study protocol (approval # 14032021, approved on 21 March 2021). Due to the non-interventional retrospective design and anonymous analyzed data, the institutional review committee waived the requirement for individual informed consent.

### 3.3. Data Acquisition

This study screened scans from patients who had visited Taibah University Dental Hospital (TUDH) and had undergone CBCT imaging between 2018 and 2022. CBCT images of patients between 16 and 72 years of age were included. Because image quality is an important factor affecting the performance of deep neural networks in computer vision tasks, poor-quality images were excluded from the collected dataset to avoid impacting the effectiveness of the applied deep learning models and their results. Therefore, images belonging to patients with a history of trauma, pathology, surgical intervention congenital syndrome, fracture, or any other foreign body (which produced artifacts in the image) were excluded from the study. Moreover, blurred images, such as those containing an ill-defined and unclear definition of bony borders, were excluded from the study. Of 890 screened scans, 500 images fulfilled the eligibility criteria. Anonymous code was given to each subject, all the data was saved in an excel sheet with a security password, and the computer was secured with a password. Notably, visualizing anatomical structures depends on technological parameters such as image resolution and CBCT reconstruction time. In this study, all CBCT scans performed in the TUDH adhered to a standardized scanning protocol with the same machine (KaVo 3D eXam; KaVo) at the same specified device settings (i.e.,120 kVp and 5 mA using a field of view of 16×13 cm, voxel size of 0.25 mm, and 26.9 s of acquisition time) as a change in any of these factors may affect the visualization of anatomical structures.

Images were reconstructed in C.S. 3D Imaging Software (Carestream Dent L.L.C., Atlanta, GA, USA). The head positions were standardized anteroposteriorly and sagittally according to a standard method. From CBCT volumes, the panoramic images were reconstructed by selecting a custom focal trough that passed through the lingual cusps of the maxillary teeth and extended posteriorly to the condyles. (Figure 2).

This conversion process is summarized in Algorithm 1.
**Algorithm 1 Conversion of CBCTimages to X-ray Images****1.**  **Input**: 3D cone beam CBCTdataset**2.**  **Output**: X-ray images**3.**  **Begin****4.**  X_ray_dataset  = [ ] // empty dataset**5.**  **for** all img in dataset **do**:
**6.**    **if** image_quality is poor **Then:****7.**       valid_dataset = RemoveImage(img)
**8.**    **else**:9.
       SaveImage( ) // save images as DICOM**10.**  RecoImage( ) // reconstruct images using C.S. 3D Imaging Software**11.**  SliceImage(img) // slice images **12.**  X_ray_dataset=SaveImg(img) // Save images as PNG format**13.** **end for****Return** X-ray images

### 3.4. Data Preprocessing and Segmentation

The X-ray images dataset obtained by applying algorithm 1 was passed to the next stage. In the data preprocessing phase, the images were standardized, and all panoramic dental X-ray images were resized to 512 × 512 pixels and normalized in the range of 0 to 1.

After preprocessing, the datasets were randomized and divided into 70% training, 20% validation, and 10% test data, as shown in Figure 1. Since there are two separate experiments, the training set was also used to provide experience for segmentation models regarding teeth segments. For this purpose, only 120 images were used for segmentation, while the remaining data were used to evaluate the segmentation models.

### 3.5. Detection and Classification Using the Pretrained DL Models

A total of six pretrained CNN models were used in this study, which include AlexNet, VGG16, VGG19, ResNet50, DenseNet169, and MobileNetV3. Each of these models is described in detail in the following subsections.

#### 3.5.1. AlexNet

AlexNet is a CNN model proposed by Krizhevsky et al. [26]. It has eight layers with learnable parameters, including five convolutional layers, three fully connected layers, and three max-pooling layers that are connected after the first, second, and fifth convolutional layers, respectively. As an activation function, AlexNet model uses the Relu activation function in all its layers except the output layer. The original AlexNet architecture and its configuration settings are shown in Figure 3 and Table 1, respectively.

#### 3.5.2. VGG16 and VGG19

VGG16 is also a type of CNN model that consists of 21 layers. Out of these, there are only 16 weight layers with learnable parameters. Precisely, the model consists of 13 convolutional layers, 3 fully connected layers, and 5 max pooling layers [27]. The VGG16 model uses small convolutional filters (3 × 3) with stride 1. In addition, the same padding and max pool layer of a 2 × 2 filter of stride 2 are used.

This network has about 138 million parameters, which is quite large. The original VGG16 architecture and its configuration settings are depicted in Figure 4.

VGG19 is a variant of the VGG16 model in which three new convolutional layers were added up to the previous VGG16 model. There are 19 layers (16 convolution layers and 3 fully connected), 5 max-pooling layers, and 1 SoftMax layer. Figure 5 shows the VGG19 architecture.

#### 3.5.3. ResNet50

ResNet50 is a variant of the ResNet model, which has 48 convolution layers along with 1 MaxPool and 1 average pool layer. It has 3.8 × 10^9^ floating point operations. Since the vanishing gradient problem is common in CNN-based models during backpropagation, the ResNet-based model uses a “skip connection” to overcome such a problem, as shown in Figure 6.

#### 3.5.4. DenseNet169

A dense convolutional network (DenseNet) is an expansion of the Residual CNN (ResNet) architecture. Indeed, there are several DenseNets such as, DenseNet121, DenseNet169, and DenseNet201. In this paper, the DenseNet169 was used. As indicated in its name, the architecture of this model contains 169 layers. The reason for this choice is due to the ability of DenseNet169 to handle the vanishing gradient problem compared to the other versions [28]. In addition, it has a minimal number of parameters. In contrast to ResNet and other convolution neural networks, DenseNet-based models provide an immediate connection between each layer and all descendant layers, as shown in Figure 7.

#### 3.5.5. MobileNetV3

The MobileNetV3 proposed by Howard et al. [29] is an enhanced version of the original MobileNetV1 and MobileNetV2. The MobileNetV3 uses a new network architecture search algorithm called the NetAdapt algorithm for two reasons: (i) to search for the best kernel size and (ii) to optimize the MobileNet architecture by fulfilling the hardware platforms in terms of size, performance, and latency. The MobileNetV3 processes a new nonlinearity version of a sigmoid function called hard swish (h-swish), which is used, on the one hand, to reduce the overall model complexity and size and, on the other hand, to minimize the number of training parameters:(1)h−swish(x)=x ⋅σ(x) where,
(2)$$\sigma \left(x\right)=\frac{ReLU\;6\left(x+3\right)}{6}$$

### 3.6. Experimental Setup

The proposed approach is implemented on a machine that meets the following specifications: Intel(R) Core (T.M.) i9-10980HK CPU @ 2.40 GHz 3.10 GHz processor and a 32 GB RAM running Windows 11 with an NVIDIA GeForce MX graphics card. The Jupyter Notebook provided with Anaconda distribution [30] and Keras 2.10.0 library with TensorFlow backend was used for implementing all the proposed DL models using Python 3.9 [31]. During the segmentation phase, a 10-fold cross-validation technique was applied, and the sigmoid output of the network was directly binarized. In addition, the weights were randomly initialized using a truncated normal distribution centered at zero. Then, the weights were optimized using the Adaptive Moment Estimation (ADAM) optimizer over 250 epochs with a batch size of four. We set the learning rate to 0.001. In terms of data augmentation, various combinations of horizontal flipping, vertical flipping, and rotations were applied. Table 2 shows the hyperparameters used for model training at the segmentation and classification phase.

### 3.7. Evaluation Metrics

Several metrics were used to evaluate the proposed models’ performance depending on the model type [27]. Among all metrics, accuracy, precision, recall, and F1-score were most used. These metrics are denoted by the following Equations (3)–(6) as follows:(3)$$Accuracy=\frac{TP+TN}{TP+TN+FP+FN}$$
(4)$$Precision=\frac{TP}{TP+FP}$$
(5)$$Recall=\frac{TN}{TN+FP}$$
(6)$$F1\;Score=2\times \frac{\;Precision\times Recall}{Precision+Recall}$$

In addition, we use the Matthews correlation coefficient (MCC) to measure how predictions match labels. The MCC is defined as follows:(7)$$CC=\frac{TP/n-\bar{S}\bar{P}}{\sqrt{\bar{S}\bar{P}\left(1-\bar{S}\right)(1-\bar{P})}}$$
where $\bar{S}$ is the class label, $\bar{P}$ is the prediction of the data, n is the total number of observations, and TP is “True Positives”.

## 4. Experimental Results and Discussion

This section summarizes the findings of the experiments performed on the dataset by applying the U-Net segmentation model and without segmentation.

### 4.1. Results Using Teeth Segmentation

The U-net model was trained and validated on a dataset of 120 images, and then the test set was used to evaluate the overall performance. Table 3 shows the performance results regarding the training, validation, and testing set. The results showed that the model was stable and achieved an accuracy of 90.81% for the testing set. In addition, it is notable that the training and validation results are closed, which means the model does not fall into the overfitting problem. In addition, this is a good sign that the model was generalized fine.

The confusion matrix presented in Table 4 and Figure 8 show that the U-Net model can predict the missing teeth (class 1) perfectly.

### 4.2. Classification Results of Pretrained DL Models

This section presents the performance of the pretrained DL models (AlexNet, VGG16, VGG19, ResNet50, DenseNet169, and MobileNetV3) using various performance metrics. It is important to mention that the obtained images from the segmentation stage were fed to the DL classifiers. Table 5 shows the performance of the pretrained DL models where the U-Net model was used as a backend segmentation tool. Table 5 also gives a fair idea about the superiority of DenseNet169 implementation results. It shows that the AlexNet model produced the worst results, with an accuracy of 67.75%, then MobileNetV3 with an accuracy of 82.50%, while VGG16 and ResNet50 yielded an accuracy of 90%.

Since the main goal of this paper is to help dentists in future planning for placing dental implants, we need models that can predict the missing position of teeth more accurately. Thus, the highest F1-score of class 1 (the missing teeth class) indicates that the model detects the position of missing teeth more accurately. Figure 9 shows the performance of the pretrained models in terms of precision, recall, and F1-score. The recall of the models shows how many times the model was able to detect a specific category. The DenseNet169 is still the winner among all the other models, with recall of 0.98. AlexNet shows an interesting behavior with a recall of 0.97. Similarly, MobileNetV3 gives a recall of 0.97. The worst value is obtained with VGG16, with a recall of 0.93. In terms of precision, where the goal is to determine how good the model is at predicting the missing teeth class, the DenseNet169 gives the best value with a precision of 0.89, while AlexNet gives the worst value with a precision of 0.61. The performance of VGG16 is closer in terms of precision to the performance of the DenseNet169 model, with a value of 0.88.

Regarding the normal teeth class, the performance of the proposed DL models in terms of precision was above 0.90. Figure 10 shows the superiority of the DenseNet169 with a precision of 0.98, followed by MobileNetV3 with a precision of 0.95 and VGG19 and ResNet50 with a precision of 0.94 each. VGG16 and AlexNet followed with a precision of 0.93 and 0.92, respectively. Figure 10 also shows that the AlexNet model yielded the worst performance in terms of recall with a value of 0.38. In general, the recall values were between 0.68 for MobileNetV3 and 0.88 for the DenseNet169 model.

The obtained results of these deep learning models (such as DenseNet169) show that they overcome the performance of the models proposed in the previous studies. For instance, the precision values of the detected missing tooth regions were 95%, 90%, and 59.09% in [6,18,22], respectively.

Table 6 shows the MCC values for all models. The higher the value, the higher agreement between actuals and predictions. In our case, DenseNet169 is the best model compared to the others. Figure 11 shows the normalized confusion matrices for all the proposed models.

### 4.3. Classification without Segmentation

This subsection summarizes the findings of the experiments performed on the dataset without applying any segmentation model. The results in Table 7 show that VGG16 overcomes all the remaining DL models. It achieved an accuracy of 79.17%. The performance of DenseNet169 and VGG19 is the same in terms of accuracy, recall, and F1-score. However, the DenseNet169 exceeds VGG19 in terms of precision, with a value of 0.81. The AlexNet model gives the worst performance, with an accuracy of 52.50%.

Figure 12 shows the performance of the pretrained models with respect to the missing teeth class in terms of precision, recall, and F1-score. It shows that DenseNet169, VGG16, and VGG19 achieved F1-scores of 0.81, whereas their precision and recall are different. The ResNet50 model produced the best recall results, with a value of 0.98, whereas AlexNet provided the worst results.

Figure 13 shows the performance of the DL models when the class of normal teeth is considered. Among all the models, ResNet50 produced the worst results with F1-score of 0.15 and recall of 0.08. DenseNet169 had a precision of 90% and recall of 63%. VGG16 achieved an F1-score of 77%.

The Matthews correlation coefficient (MCC) shows that AlexNet, ResNet50, and MobilenetV3 failed to predict the class label, whereas DenseNet169, VGG19, and VGG19 were, to some extent, capable of predicting the class label as shown in Table 8.

### 4.4. Statistical Analysis

The previous sections showed that DenseNetV3 could predict the teeth’s class label more accurately than the other models. As a result, we chose it as the recommended approach for predicting missing teeth. However, to show that the choice is statistically significant, the statistical significance of the results was measured using the Mann–Whitney–Wilcoxon test and the Wilcoxon signed-rank test. For this purpose, we formulated the null hypothesis, h0, as follows: the metrics scores of DenseNet169 were achieved after applying the U-Net model as a segmentation backend, and the metrics scores without segmentation are the same. The p values from the Mann–Whitney–Wilcoxon test suggest that the null hypothesis can be rejected due to the significance level below 0.05, and the null hypothesis can be dismissed since the improvement is significant (See Table 9).

## 5. Conclusions and Future Research Directions

Several pretrained deep learning models were implemented in this study, including AlexNet, VGG16, VGG19, ResNet50, DenseNet169, and MobileNetV3, and their performance in detecting and classifying the missing tooth regions was also investigated in terms of different performance evaluation metrics. In addition, two types of feature extraction were investigated. The feature extraction was based on the U-Net model, and the other type used the bare image without segmentation. A binary classification-based dataset was obtained to facilitate the process of identification of the missing teeth bone area. For future planning and implanting of the predicted missing tooth areas, a domain expert verified the resulting dataset manually.

The experimental results showed that the segmentation process played an essential role in enhancing the performance of the pretrained DL models. Among the proposed DL models, the finding showed the superiority of DenseNet169, with an F1-score of 0.94 for the missing teeth class and 0.93 for the class of normal teeth class. The use of this model may represent a promising time-saving tool serving dental implantologists with a first step toward automated dental implant planning. In addition, this study also constructed a ground-truth dataset for tooth instance segmentation and missing tooth region detection in panoramic radiographic images which were officially obtained, sliced, and extracted and reviewed by an expert with more than 10 years of professional experience in dental implant planning. The domain expert manually annotated the X-ray images to obtain ground-truth masks using the well-known annotation tool called Label Studio (https://labelstud.io/, accessed on 3 October 2022). An improved algorithm and more data will make implant placement more automated in the future.

## Figures and Tables

**Figure 1 ijerph-19-15414-f001:**
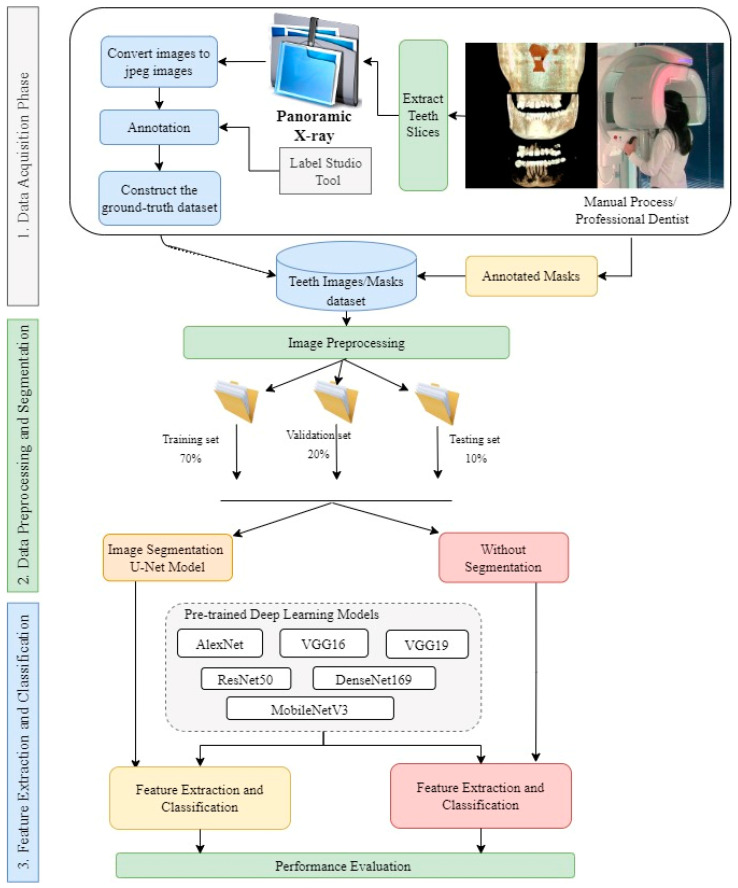
The study design methodology.

**Figure 2 ijerph-19-15414-f002:**
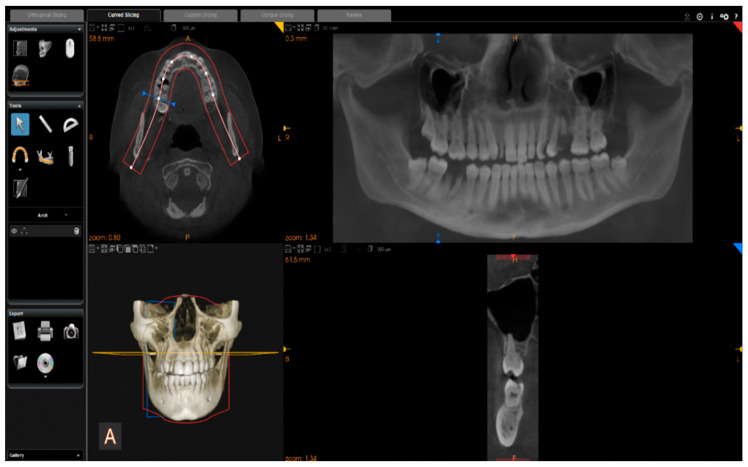
Reconstruction of panoramic images from CBCT volumes.

**Figure 3 ijerph-19-15414-f003:**
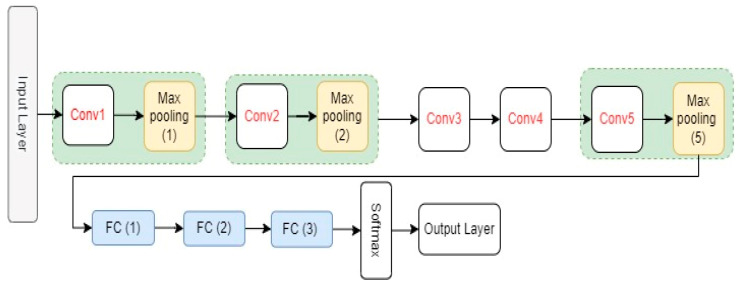
The architectures of AlexNet.

**Figure 4 ijerph-19-15414-f004:**
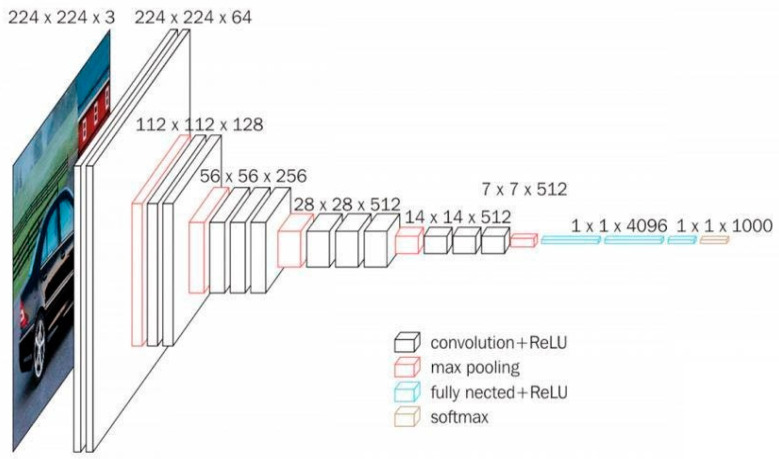
The original VGG16 architecture.

**Figure 5 ijerph-19-15414-f005:**
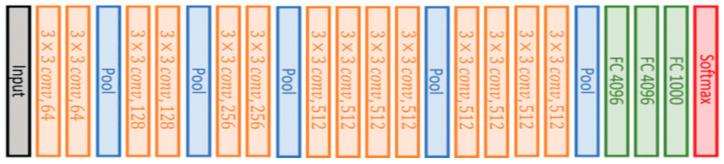
VGG19 original architecture.

**Figure 6 ijerph-19-15414-f006:**
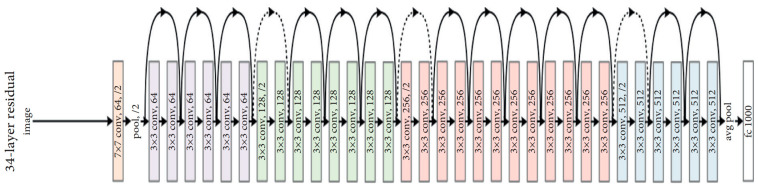
Architecture of the ResNet50 model, the black curved arrows show “skip connection” among the layers.

**Figure 7 ijerph-19-15414-f007:**
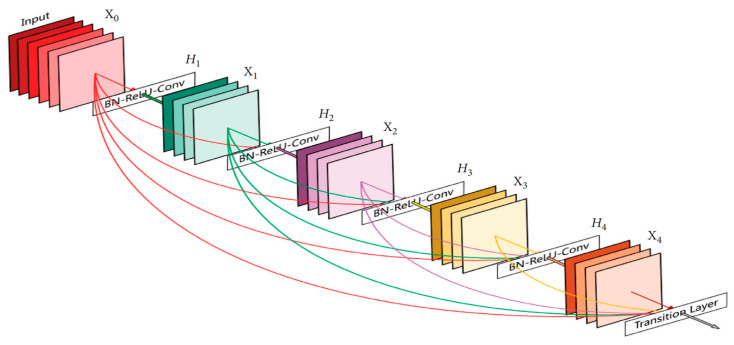
Architecture of DenseNet model. Each layer connects all the subsequent network layers.

**Figure 8 ijerph-19-15414-f008:**
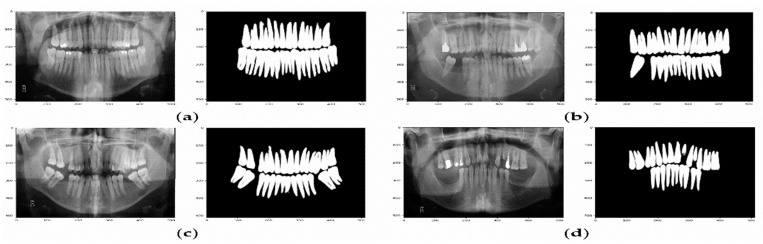
Example of predicted masks using U-Net model. (**a**) The X-ray image and the predicted binary mask for normal teeth. (**b**–**d**) Various X-ray images and their predicted binary masks for missing teeth.

**Figure 9 ijerph-19-15414-f009:**
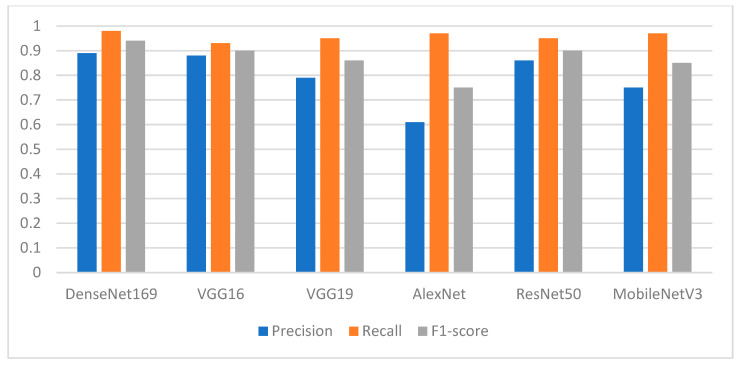
Performance of the models in terms of precision, recall, and F1-score with respect to class 1 (missing teeth class).

**Figure 10 ijerph-19-15414-f010:**
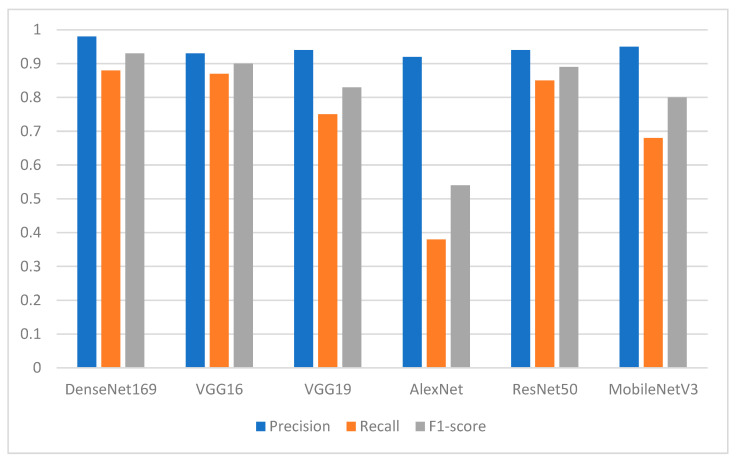
Performance of the models in terms of precision, recall, and F1-score with respect to class 0 (class of normal teeth).

**Figure 11 ijerph-19-15414-f011:**
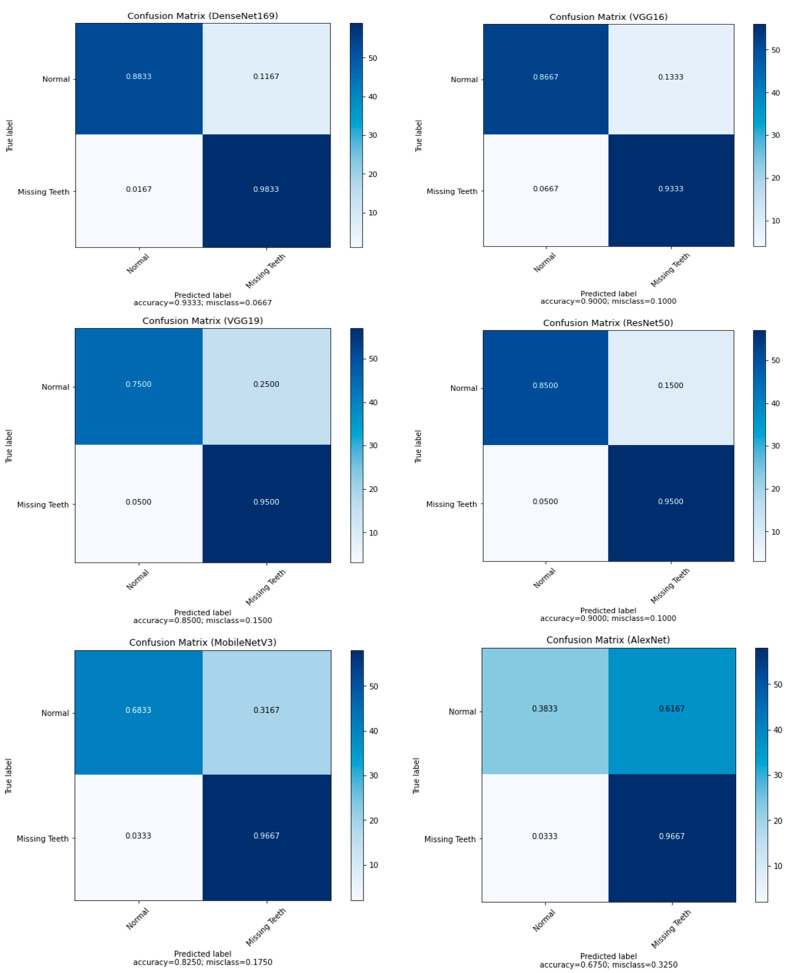
Confusion matrix for all pretrained models.

**Figure 12 ijerph-19-15414-f012:**
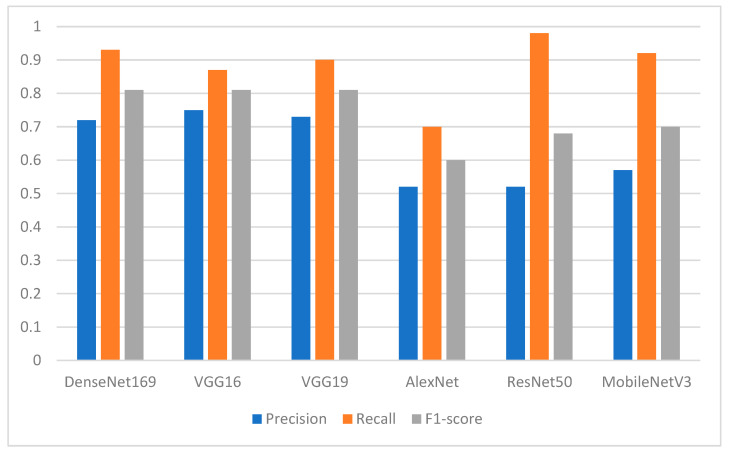
Performance of the models in terms of precision, recall, and F1-score with respect to class 1 (class of missing teeth).

**Figure 13 ijerph-19-15414-f013:**
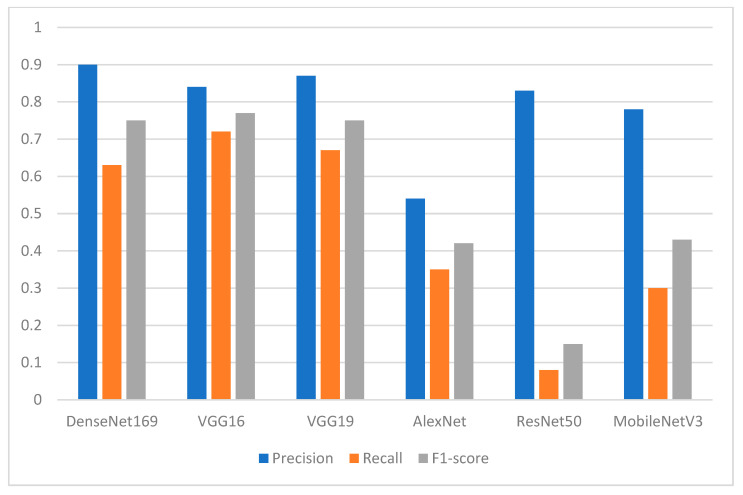
Performance of the models in terms of precision, recall, and F1-score concerning class 0 (class of normal teeth).

**Table 1 ijerph-19-15414-t001:** The original AlexNet model architecture.

Layer	# of Filters/Neurons	Filter Size	Stride	Padding	Size of Feature Map	Activation Function
Input Layer	-	-	-	-	227 × 227 × 3	-
Conv (1)	96	11 × 11	4	-	55 × 55 × 96	ReLu
Conv (2)	256	5 × 5	1	2	27 × 27 × 256	
Conv (3)	384	3 × 3	1	1	13 × 13 × 384	
Conv (4)	384	3 × 3	1	1	13 × 13 × 384	
Conv (5)	256	3 × 3	1	1	13 × 13 × 256	
Max pooling (1)	-	3 × 3	2	-	27 × 27 × 96	-
Max pooling (2)	-	3 × 3	2	-	13 × 13 × 256	
Max pooling (3)	-	3 × 3	2	-	6 × 6 × 256	
Dropout 1	Rate = 0.5	-	-	-	6 × 6 × 256	-
Dropout 2	Rate = 0.5	-	-	-	4096	-
Fully Connected (1)	-	-	-	-	4096	ReLu
Fully Connected (2)	-	-	-	-	4096	ReLu
Fully Connected (3)	-	-	-	-	1000	Softmax

**Table 2 ijerph-19-15414-t002:** The used hyperparameters for the model configuration.

Model Training Hyperparameters	Values
Images Size	512 × 512 × 3
Epochs	250
Optimizer	ADAM
Learning rate	1 × 10^−3^
Activation function	Sigmoid
Batch Size	4
Loss function	Binary Cross-entropy

**Table 3 ijerph-19-15414-t003:** U-Net best performing training, validation, and testing results.

Model	Accuracy	Precision	Recall	F1-Score	Loss
U-Net Training	93.40	0.99	0.97	0.98	0.0118
U-Net Validation	92.92	0.93	0.98	0.96	0.0116
U-Net Testing	90.81	0.96	0.97	0.97	0.0580

**Table 4 ijerph-19-15414-t004:** Confusion matrix of U-net model.

	Precision	Recall	F1-Score
Class 0 (Normal)	0.93	0.98	0.96
Class 1 (Missing teeth)	0.99	0.97	0.98
Macro avg.	0.96	0.97	0.97
Weighted avg.	0.97	0.97	0.97

**Table 5 ijerph-19-15414-t005:** Performance results of the deployed pretrained DL models with segmentation.

Model	Accuracy	Precision	Recall	F1-Score
DenseNet169	93.33	0.94	0.93	0.93
VGG16	90.00	0.90	0.90	0.90
VGG19	85.00	0.86	0.85	0.85
AlexNet	67.75	0.77	0.68	0.64
ResNet50	90.00	0.90	0.90	0.90
MobileNetV3	82.50	0.85	0.82	0.82

**Table 6 ijerph-19-15414-t006:** MCC performance results of the deployed pretrained DL models with U-net segmentation.

Model	MCC
DenseNet169	0.8710
VGG16	0.8351
VGG19	0.7144
AlexNet	0.4309
ResNet50	0.8040
MobileNetV3	0.6778

**Table 7 ijerph-19-15414-t007:** Performance results of the deployed pretrained DL models.

Model	Accuracy	Precision	Recall	F1-Score
DenseNet169	78.33	0.81	0.78	0.78
VGG16	79.17	0.80	0.79	0.79
VGG19	78.33	0.80	0.78	0.78
AlexNet	52.50	0.53	0.52	0.51
ResNet50	61.67	0.68	0.53	0.41
MobileNetV3	60.83	0.67	0.61	0.57

**Table 8 ijerph-19-15414-t008:** MCC performance results of the deployed pretrained DL models.

Model	MCC
DenseNet169	0.5940
VGG16	0.5900
VGG19	0.5828
AlexNet	0.0534
ResNet50	0.1529
MobileNetV3	0.2752

**Table 9 ijerph-19-15414-t009:** The reported *p* values for DenseNet169.

Test	Accuracy	Precision	Recall	F1-Score
Mann–Whitney–Wilcoxon test (Wilcoxon rank sum test)	0.01453	0.01453	0.01453	0.01453
Wilcoxon signed-rank test	0.04718	0.04718	0.04718	0.04718

## Data Availability

Data is available upon request from authors.

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
