# Peer review of "Enhanced Tooth Region Detection Using Pretrained Deep Learning Models"

_ijerph, 2022, doi:10.3390/ijerph192215414_

Round 1

Reviewer 1 Report

The study has applied AI (5 DL models) for detection and classification of teeth regions in dataset segmented from CBCT images.

My major concern about the study is the design and clinical relevance: for detection of the edentulous areas, a large (16 x 13 cm) FOV is needed herein. In practice, you will identify the area needed for implantation simply from a clinical examination, and then scan only this specific area (adhering to the ALARA principle to reduce the radiation dose).

The teeth regions detection is only the first/"simple" step towards automated implant planning - and not a major step as claimed by the authors in the conclusion. Furthermore, detection and classification of teeth / edentulous areas may be applied to other fields of dentistry. Therefore, I suggest avoiding “for dental implant planning” in the Title.

Abstract: The abbrev. CNN must be explained (and not only in main text page 3 – should here be explained the first time: line 130).

DenseNet169 has been spelled in different ways (also in the main text; and in Fig 1: DenseNet 161).

Introduction: References 1 and 7 seem not relevant to cite in this context. Other articles should be cited instead.

Generally, the manuscript should be restructured. The aim is described two times (page 2, line 67, page 4, line 174). The Intro and Related works segments should be fused (several repetitions), 4.1 Experimental design should be described in M&M.

The paper’s organization does not have to be explained in the manuscript (e.g. page 2, line 93).

M&M: Was the professional dentist producing the ground-truth a radiologist / dentist experienced in CBCT segmentation specifically?

CBCT throughout the manuscript (and not cone beam C.T. images some places – or be consistent).

Poor-quality images have been excluded. Please discuss that this may have an impact on the results.

Results: Page 12, line 358: “implanting teeth” should be replaced with “placing dental implants”.

Conclusions: “major step” should be replaced with “first step“.

Author Response

Dear reviewer, we would like to express our sincere gratitude and appreciation for your valuable time and efforts.

Below are our responses to your valuable comments.

The study has applied AI (5 DL models) for detection and classification of teeth regions in dataset segmented from CBCT images.

  • My major concern about the study is the design and clinical relevance: for detection of the edentulous areas, a large (16 x 13 cm) FOV is needed herein. In practice, you will identify the area needed for implantation simply from a clinical examination, and then scan only this specific area (adhering to the ALARA principle to reduce the radiation dose).

Response: thank you for your valuable comment. We concur with you regarding this point. In our experiment, we retrospectively used CBCT images already taken for cases that required additional three-dimensional imaging evaluation for diagnosis and treatment purposes. In practice, the case indicated for implant placement required a CBCT scan to avoid violating the vital anatomical structure. As outlined in the conclusion, the use of this model represents a promising time-saving tool serving dental implantologist with the first step toward automated dental implant planning. This work is part of a large project to reach a fully automated dental implant planning process.

  • The teeth regions detection is only the first/"simple" step towards automated implant planning - and not a major step as claimed by the authors in the conclusion. Furthermore, detection and classification of teeth / edentulous areas may be applied to other fields of dentistry. Therefore, I suggest avoiding “for dental implant planning” in the Title.

Response: amended as requested.

  • Abstract: The abbrev. CNN must be explained (and not only in the main text on page 3 – it should here be explained the first time: line 130).

Response: amended as requested.

  • DenseNet169 has been spelled in different ways (also in the main text; and in Fig 1: DenseNet 161).

Response: A dense convolutional network (DenseNet) is an expansion of the Residual CNN (ResNet) architecture. In deeded, there are several DenseNet such as, DenseNet121, DenseNet169, DenseNet201. In this paper, the DenseNet169 was used. Figure 1 has been revised to use DenseNet169.

  • Introduction: References 1 and 7 seem not relevant to cite in this context. Other articles should be cited instead.

Response: amended as requested.

  • Generally, the manuscript should be restructured. The aim is described two times (page 2, line 67, page 4, line 174). The Intro and Related works segments should be fused (several repetitions), 4.1 Experimental design should be described in M&M.

Response: The aim of the paper has been described in page 2, and revised in page 4.

The intro has been revised to eliminate repetitions with the Related works. More descriptions of A.I. technologies and applications were detailed in the Related works only, while the intro focused on defining the main research problem and proposing the solution. In addition, the paper contribusions were moved to the discussion. The paper’s organization was removed from the intro.

The experimental design is described in the M&M, as suggested.

The paper’s organization does not have to be explained in the manuscript (e.g. page 2, line 93).

Response: The paper’s organization is removed as suggested.

  • M&M: Was the professional dentist producing the ground-truth a radiologist / dentist experienced in CBCT segmentation specifically?

Response: As mentioned in the Introduction and added to M&M: an expert with more than 10 years of professional experience in dental implant planning reviewed all images.

  • CBCT throughout the manuscript (and not cone beam C.T. images some places – or be consistent).

Response: amended as requested.

  • Poor-quality images have been excluded. Please discuss that this may have an impact on the results.

Response: amended as suggested. See page 5, line 212

  • Results: Page 12, line 358: “implanting teeth” should be replaced with “placing dental implants”.

Response: amended as suggested.

  • Conclusions: “major step” should be replaced with “first step“.

Response: amended as suggested.

Reviewer 2 Report

Mohammed Al-Sarem et al. applied several pre-trained deep learning models for enhancing  3D missing teeth area planning, including AlexNet, VGG16, VGG19, ResNet50, DensNet169, and Mo-bileNetV3 models. And they found DenseNet169 was superior with F1-score of 0.94 for the 442 missing teeth class and 0.93 for the class of normal teeth class. The paper is well-written and of certain clinical significance.

Minor suggestions

Abstract

1. Line 14-19 too much background information in the abstract. Please shorten these sentences.

2. They should report how many cases/images were included in the research

Introduction

3. Line 69-92 The author talked the main contribution of this research, which might be better put in the discussion section.

4. line 73-75. The ethical approval should be put in the methods section.

5. Line 93-99 This paragraph might not be necessary. As there were sub headings for each section.

Author Response

Dear reviewer, we would like to express our sincere gratitude and appreciation for your valuable time and efforts.

Below are our responses to your valuable comments.

The paper is well-written and of certain clinical significance.

Response: Thank you

Minor suggestions

Abstract

  1. Line 14-19 too much background information in the abstract. Please shorten these sentences.

Response: The background information was shortened.

  1. They should report how many cases/images were included in the research

Response: As described in (3.3. Data Acquisition) section, 500 images fulfilled the eligibility criteria and used in this study.

Introduction

  1. Line 69-92 The author talked the main contribution of this research, which might be better put in the discussion section.

Response: The main contribusions were moved to the discussion..

  1. line 73-75. The ethical approval should be put in the methods section.

Response: The ethics statement was included in section 3. Material and Methods. 

  1. Line 93-99 This paragraph might not be necessary. As there were sub headings for each section.

Response: The paper organization paragraph has been removed.   

Round 2

Reviewer 1 Report

Most of my points have been addressed sufficiently. However, there are still few things that must be revised.

In the abstract, there are still different spellings of DenseNet169: DensNet, Densenet, DenseNet

The possible impact of excluding poor-quality images on the results has not been discussed as requested? I do not find it on page 5, line 212 as mentioned in the response from the authors. Please dicuss briefly

Author Response

Dear reviewer 

We appreciate your time and efforts. 

Our responses to your valuable comments are listed below 

In the abstract, there are still different spellings of DenseNet169: DensNet, Densenet, DenseNet

Response: the text has been amended.

-The possible impact of excluding poor-quality images on the results has not been discussed as requested? I do not find it on page 5, line 212 as mentioned in the response from the authors. Please dicuss briefly

Response: Thank you very much for your comment. The following statements were added to 3.3. Data Acquisition  page 5-6 (lines 184-200)

Because image quality is an important factor affecting the performance of deep neural network in computer vision tasks, poor-quality images were excluded from the collected dataset to avoid impacting the effectiveness of the applied deep learning models and their results. Therefore images belonging to patients with a history of trauma, pathology, surgical intervention congenital syndrome, fracture, or any other foreign body (which produced artifacts in the image) were excluded from the study. Moreover, Blurred images, such as those containing an ill-defined and unclear definition of bony borders, were excluded from the study. Of 890 screened scans, 500 images fulfilled the eligibility criteria. Anonymous code was given to each subject, all the data was saved in an excel sheet with a security password, and the computer was secured with a password. Notably, visualizing anatomical structures depends on technological parameters such as image resolution and CBCT reconstruction time. In this study, all CBCT scans were performed in the TUDH adhered to a standardized scanning protocol with the same machine (KaVo 3D eXam; KaVo) at the same specified device settings (i.e.,120 kVp and 5 mA using a field of view of 16×13 cm, voxel size of 0.25 mm, and 26.9 seconds of acquisition time); As a change in any of these factors may affect the visualization of anatomical structures.